# Parallel Strategy Increases the Thermostability and Activity of Glutamate Decarboxylase

**DOI:** 10.3390/molecules25030690

**Published:** 2020-02-06

**Authors:** Qing-Fei Zhang, Sheng Hu, Wei-Rui Zhao, Jun Huang, Jia-Qi Mei, Le-He Mei

**Affiliations:** 1College of Pharmaceutical Science, Zhejiang University of Technology, Hangzhou 310014, China; 2111707075@zjut.edu.cn; 2Department of Biological and Pharmaceutical Engineering, Ningbo Institute of Technology, Zhejiang University, Ningbo 315100, China; genegun@zju.edu.cn (S.H.); zwr166@sohu.com (W.-R.Z.); 3School of Biological and Chemical Engineering, Zhejiang University of Science and Technology, Hangzhou 310023, China; huangjun@zust.edu.cn; 4Hangzhou Zhongmei Huadong Pharmaceutical Co. Ltd., Hangzhou 31011, China; meijiaqi@eastchinapharm.com

**Keywords:** glutamate decarboxylase, rational design, sequential analysis, free energy calculation, thermostability

## Abstract

Glutamate decarboxylase (GAD; EC 4.1.1.15) is a unique pyridoxal 5-phosphate (PLP)-dependent enzyme that specifically catalyzes the decarboxylation of L-glutamic acid to produce γ-aminobutyric acid (GABA), which exhibits several well-known physiological functions. However, glutamate decarboxylase from different sources has the common problem of poor thermostability that affects its application in industry. In this study, a parallel strategy comprising sequential analysis and free energy calculation was applied to identify critical amino acid sites affecting thermostability of GAD and select proper mutation contributing to improve structure rigidity of the enzyme. Two mutant enzymes, D203E and S325A, with higher thermostability were obtained, and their semi-inactivation temperature (T_50_^15^) values were 2.3 °C and 1.4 °C higher than the corresponding value of the wild-type enzyme (WT), respectively. Moreover, the mutant, S325A, exhibited enhanced activity compared to the wild type, with a 1.67-fold increase. The parallel strategy presented in this work proved to be an efficient tool for the reinforcement of protein thermostability.

## 1. Introduction

Glutamate decarboxylase (GAD; EC 4.1.1.15) is a highly efficient enzyme that specifically catalyzes the decarboxylation of L-glutamic acid to produce γ-aminobutyric acid (GABA) in the presence of pyridoxal-5-phosphate (PLP). GAD was first discovered in mammalian brain tissue in 1951, and Paul Y. Sze reported cellular regulation of GAD which has provided important information for understanding GABA-ergic neurons and their functions [1,2]. GABA exhibits several well-known physiological functions in humans, such as induction of hypotensive effects, anticonvulsant and anti-depression effects, the promotion of hormone secretion, and protection of the liver and kidney [3,4,5,6]. Due to the safety and environmental friendliness of lactic acid bacteria (LAB), LAB has been widely used in the fermentation of GABA [7]. We have cloned the GAD with high activity from *Lactobacillus brevis CGMCC NO.1306* and investigated its application in GABA biosynthesis [8]. However, the enzyme is easily inactivated at 50 °C or above, which hampered its practical utilization. Because there is a lack of alternative thermophilic homologous enzymes with competitive catalytic activity, the creation of some engineered GAD with higher heat resistance and good activity can be useful for massive biosynthesis of GABA.

Engineering proteins for stability is an exciting and challenging field since it is critical for the industrial application of enzymes. In 1991, France H. Arnold established an evolutionary approach consisting of multiple steps of random mutagenesis and screening to enhance enzyme activity in organic media [9]. Subsequently, a theory that mimics the natural evolution by sequential accumulation of random mutations that can be used to improve the tolerance of enzymes for non-natural environments was proposed [10]. Finally, along with the invention of DNA shuffling, Willem P.C. Stemmer firstly proposed the concept of directed evolution in vitro of enzyme and greatly promoted the development of protein engineering [11]. With the deepening research on protein structures, a large number of proteins with different thermostabilities have been found and the mechanisms of heat-resistance of these proteins have been understood better [12,13,14,15], which makes people look forward to using more rational design strategies to improve protein thermostability.

The rational strategies for protein thermostability engineering can be divided into two main classes. One strategy was based on sequence information, which identified critical regions or amino acid residues by sequence alignment. Pantoliano and coworkers first suggested that sequence homology consensus may help to stabilize proteins and this approach has been applied by other researchers [16,17,18]. One typical case of this method is Lehmann et al. who constructed a synthetic gene encoding all consensus amino acids, and the corresponding protein was 15–26 °C more thermostable than any of its parents [19]. Because proteins from extremophiles can usually tolerate high temperatures better than homologous proteins from mesophiles, sequence alignment between mesophilic proteins and their homologous thermophilic proteins also facilitated to identify which residues should be mutated and how to replace them in order to make the mesophilic proteins more thermoresistant [20,21]. The other strategy, however, was based on structural information, which determined flexible regions of target proteins and strengthened these regions. For example, molecular dynamics (MD) simulations were used to identify flexible regions in haloalkane dehalogenase, and its stability was enhanced by introduction of a disulfide bond [22]. Introducing proline is another reliable approach to stabilizing proteins, because proline is the least flexible amino acid, and its ring limits its backbone conformations [23,24]. Recently, statistical analysis of protein structure–function relationships was also used to predict amino acid interactions and help to guide molecular engineering [25,26].

Several strategies have been applied to enhance the thermostability of GAD. Jun et al. improved the thermostability of GadB (*Escherichia coli*) through structural optimization of its N-terminal interdomain. The triple mutant (Gln5Asp/Val6Ile/Thr7Glu) showed 7.7 °C increases in the melting temperature (T_m_) compared to the wild type [27]. Before the crystal structure of GAD1407 was obtained, a Ramachandran plot analysis of a three-dimensional simulation structure was conducted to identify the unstable residue site, and half-life of the corresponding mutant K413A at 50 °C was 2.1-fold in comparison with the wild type [28]. Fang et al. introduced proline at 13 residue positions in GAD after homologous sequence alignment between GAD from *Lactobacillus brevis* CGMCC No.1306 and from *Thermococcus kodakarensis*, and a mutant enzyme G364P with higher thermostability was obtained [29]. In this study, we applied a similar strategy to engineering GAD from *Lactobacillus brevis* CGMCC NO.1306 but made it simpler by comparing sequence and structure of GAD with homologous thermophilic enzymes to identify key sites influencing stability [30]. Compared to the above methods, accuracy of such homologous sequence alignment based on the correlation between the enzyme’s structure and characteristics may be higher. In addition, we also applied Gibbs energy simulation to improve the efficiency of screening promising mutants; that is, a proper replacement of amino acid substituent contributing to structure rigidity should lower the Gibbs energy of protein folding [20].

## 2. Results

### 2.1. Analysis of the Mutant GAD by Sequence Alignment

GAD is a PLP-dependent decarboxylase, an imine linkage between PLP and the active site lysine (K279) ensuring the PLP is properly oriented at the active site for the reaction with the substrate. In addition, some key residues (Ser126, Ser127, Cys168, Ile211, Ser276, His278, and Ser321) also play important roles in binding PLP cofactor inside the active site and supporting its catalytic reactivity [31]. Analysis of DNAMAN on the five thermophilic GADs revealed sites near the active center are highly similar (Figure 1a), which may be related to their similar structural and functional properties. According to the consensus approach to stabilize proteins, amino acid residues that are conserved within thermophilic proteins but absent in mesophilic proteins are promising candidates. In this way, 10 residue sites were selected, and the corresponding mutations were determined as A35P, I105M, A133G, I159L, I178V, N193D, D203E, Y204N, I291V, and S325A locations of these sites in the three-dimensional structure of GAD shown in Figure 1b. The ΔG unfold of these mutations were calculated by Fold X3.0, using a full atomic description of the structure of the GAD. The different energy terms taken into account in Fold X have been weighted using empirical data obtained from protein engineering experiments [32]. As shown in Table 1, there were five sites with ΔΔG _unfold_ < −0.5 kJ·mol^−1^, including D203E, N193D, S325A, A35P, and I105M. In order to identify the accuracy and avoid the deviation of this calculation method, we selected the first eight mutations for experimental verification.

### 2.2. Specific Activity and Kinetic Constants of GAD and its Mutants

As shown in Figure 2a, two mutants, N193D and S325A, exhibited higher specific activity than the wild type (WT): their specific activities were 36.28 U/mg (N193D) and 41.12 U/mg (S325A), respectively. Compared to the specific activity of the wild type (24.57 U/mg), the best two mutants showed a 1.48-fold and a 1.67-fold increase, respectively. We measured the inactivation profile of the above eight enzymes at 60 °C for 20 min (Figure 2b). The residual activity of D203E and S325A was 72.5% and 61.0%, respectively, which was significantly higher than that of the WT (49.7%), but the specific activity of D203E was only 61.2% of that of the wild type.

The kinetic constants of the GAD and its variants were determined by monitoring the initial reaction rates at different concentrations (10–100 mmol/L) of L-glutamate (Table 2). The obvious change in catalytic efficiency (*kcat/K_M_*) was observed for the two mutants S325A (2.57 to 4.16 s^−1^·mM^−1^) and N193D (2.57 to 4.00 s^−1^·mM^−1^), but this value decreased to 1.51 s^−1^·mM^−1^ for the mutant D203E.

### 2.3. Thermal Stability of GAD and its Mutants

The thermal inactivation profile of three mutants (N193D, S325A, and D203E) and the WT at 55 °C revealed that the residual activity of the WT and N193D at 60 °C for 20 min is about 45.10% and 49.76%, respectively (Figure 2b). But as shown in Figure 3a, the half-life (t_1/2_) of the mutant N193D at 55 °C was only 33.44 ± 0.56 min and much more rapidly inactivated than the WT. That is probably because 60 °C is a rather harsh condition for both wild enzymes and mutant enzymes; hence, the two enzymes lost their activity at a similar speed; on the other hand, 55 °C was a proper condition to differentiate thermal stability of the WT and N193D, the former of which was better than the latter, i.e., mutant N193D was quickly inactivated but the WT exhibited a slower inactivation process. In contrast, the t_1/2_ of the mutant S325A at 55 °C was 125.50 ± 3.36 min, which showed a 54.32 min increase compared to the WT 71.18 ± 5.05 min. After 140 min treatment at 55 °C, the residual relative activity of the WT was only 34.27%, while mutant D203E retained a residual relative activity of 69.96% (Figure 3b). In addition, the T_50_^15^ values for D203E and S325A mutants were 62.6 and 61.7 °C, respectively. They were increased by 2.3 and 1.4 °C compared to the WT (60.3 °C), respectively. Notably, mutants D203E and S325A displayed the largest improvement in thermal stability.

### 2.4. Molecular Dynamics Simulation of GAD and its Mutants

The root mean square deviation (RMSD) (Figure 4a) and root mean square fluctuation (RMSF) (Figure 4b) were used to analyze the thermal stability of the WT and its mutants D203E and S325A. The RMSD presented here is the global value of all carbon-alpha of the backbone of each wild-type and mutant GAD. The RMSD value of the WT and its mutants initially increased significantly. After equilibration, the average RMSD of the WT was 0.77 ± 0.04 nm, and the average RMSD of D203E and S325A was 0.51 ± 0.04 nm and 0.49 ± 0.03 nm, respectively. Because lower RMSD is related to more stable conformation, it is manifested that the mutants D203E and S325A had a smaller protein conformation shift, and the stability of the protein was improved. The mutants D203E and S325A had highly similar low RMSD, but their kinetic behavior was completely different. That is probably because although the introduction of different mutations enhanced the structural rigidity, the effects on the interaction between enzyme and substrate are different. The RMSF value reflected fluctuations of each amino acid during simulation. As shown in Figure 5b, the most flexible region of the mutants and the WT was located in the N-terminus regions of the protein. The RMSF value of S325A in the loop region near the residue 325 and the N-terminus of the protein were all lower than those of the WT.

## 3. Discussion

A parallel strategy was applied to identify amino acid sites possibly critical to the thermostability of GAD and the proper mutant likely improving structural rigidity in this study. The first strategy was a ‘consensus design’ approach and the second was a Gibbs energy simulation by Fold X3.0. Such attempts created two mutants, D203E and S325A, the T_50_^15^ values of which were 2.3 °C and 1.4 °C higher than the corresponding values of the wild-type enzyme, respectively. Moreover, the mutant, S325A exhibited enhanced specific activity compared to the wild type, with a 1.72-fold increase. Such results manifested that a combination of the two strategies was useful for thermostability reinforcement of protein. However, one exceptional mutant, N193D, declined in thermostability, although the lower ΔΔG _unfold_ compared to that of S325A made it clear that stability modification is still too complicated to be accomplished by simple methods, and more accurate methods need to be explored.

A major factor in the folding of proteins is the burying of hydro-phobic side chains [33]. Pace et al. have systematically analyzed 148 mutants related to hydrophobic interactions in 22 proteins and concluded that the contribution of hydrophobic interactions to protein stability is about 60%, while the contribution of hydrogen bonding to protein stability is about 44%, burying a –CH_2_– group on folding contributes with Δ(ΔG) values of 1.1kcal/mol to protein stability [34]. Calculation of hydrophobic interaction of the wild type and mutant S325A by online server (http://pic.mbu.iisc.ernet.in/) indicated the hydrophobic interaction of mutant enzyme S325A was increased. Figure 5 shows the predicted molecular interaction simulation of wild type and S325A around position 325 (Figure 5a,b) and wild type and N193D around position 193 (Figure 5c,d). Hydrophobic interaction between Ala325 with Met78 (4.1 Å) and Leu82 (4.1 Å) generated by replacement of serine by alanine helped to enhance tightness between the two alpha-helices. This may be an important reason for the thermostability improvement of the protein. In addition, although the amino acid site of 325 is not directly involved in the reaction with the substrate, the replacement of serine by alanine at this site possibly induced the formation of two hydrogen bonds with the carboxyl group of glutamic acid and therefore facilitated the decarboxylic reaction of the substrate. However, all above analyses are hypothetical speculations and need further exploration regarding the relationship of the structure and functions of GAD.

Although the thermostability of mutant N193D was not improved, its catalytic activity was increased 1.56-fold compared to the WT. According to the PIC online server, Asp193 was involved in an ionic bond with the neighboring His196. However, residue 193 and 203 were located on the a-helix which was far away from the active center and not directly involved in the catalytic reaction, the reason for the increase in activity and thermostability is not clear.

Rational design was often applied to predict important amino acid residue sites to improve the efficiency and feasibility of molecular engineering. With rapid advances of laboratory automation, large numbers of mutant enzymes have been generated and many impressive examples of improved enzymes were conducted [35,36]. In this study, sequential analysis and free energy calculation were combined to identify critical amino acid sites and predict proper mutation contributing to improved structural rigidity of the protein. Among the eight mutations predicted by this way, two mutants with higher thermostability had been verified, which manifested that such strategy was efficient and useful for the molecular reinforcement of protein. Compared to other strategies which have been applied to enhance the thermostability of GAD, including site-directed saturation mutagenesis [37] and introduction of proline [29], our method greatly reduced the workload of screening. We believe the parallel strategy including consensus design and free energy calculation designed in this study provides an effective way to improve thermostability of other enzymes. However, the original problem of directed evolution still inherent in this strategy is that designed mutations are not reliably predictable but have to be tested individually.

## 4. Materials and Methods

### 4.1. Strains and Plasmids

pET28(a+)-GAD1407 was constructed in our laboratory [8]. P*fu* DNA polymerase and *Dpn I* enzyme was purchased from TransGen Biotech (Beijing, China). The pET-28a (+)/*Escherichia coli* BL21 (DE3) system was used for the expression of GAD and subsequent mutants.

### 4.2. Rational Design

To improve the thermostability of GAD, yet without modifying their function, we selected the mutation sites by sequence alignment. We found five GADs with high thermostability and high catalytic activity by searching the Brenda enzyme database (Table 3). The sequence alignment was achieved by DNAMAN. Next, we performed a preliminary evaluation of the mutation sites by ΔG_unfold_. The ΔG_unfold_ of these mutations was calculated by Fold X3.0 [32], using the GAD (PDB:5GP4) structure as a reference (ΔΔG_unfold_ = ΔG_mutation_-ΔG_wild-type_). The standard settings of the software were used (T = 323 K/333K/343K, pH = 5.0, ionic strength = 0.05 M). Mutations were evaluated as stabilizing if ΔΔG _unfold_ < −0.5 kJ·mol-1. The free energy of unfolding (ΔG) of a target protein is calculated using the equation:ΔG=Wvdw⋅ΔGvdw+WsolvH⋅ΔGsolvH+WsolvP⋅ΔGsolvP+ΔGwb+ΔGhbond+ΔGel+ΔGKon+Wmc⋅T⋅ΔSmc+Wsc⋅T⋅ΔSsc[32]

### 4.3. Construction of Mutants

The site-directed mutagenesis primers were designed according to the GAD gene of *Lb. brevis CGMCC No.1306*.The plasmid containing the GAD1407 gene was used as a template for site-directed PCR amplification. Amplification was performed using the following temperature settings: 5 min at 94 °C; 30 cycles of 30 s at 94 °C, 30 s at 58 °C, and 4 min at 72 °C; final 5 min extension at 72 °C. PCR products were digested with *Dpn I* for 1 h at 37 °C to remove the parent plasmid and then transformed into *E.coli* BL21 (DE3) competent cells. The solution was cultured at 37 °C for 1 h. The cultures were applied to a Luria-Bertani (LB) medium plate containing 50 μg/mL kanamycin and cultured overnight at 37 °C. DNA sequencing was completed by Sangon Biotech.

### 4.4. Enzyme Expression and Purification

The wild-type and mutant enzymes were inoculated in a 100 mL LB medium flask with 50 μg/mL kanamycin and cultured at 37 °C with shaking. When optical density at 600 nm (OD600) of the culture reached 0.6−0.8, Isopropyl-beta-D-thiogalactopyranoside (IPTG) (final concentration 100 mg/mL) was added to induce protein expression (28 °C for 8 h). The cells were collected by centrifugation (4000 rpm) at 4 °C for 10 min and suspended by phosphate-buffered saline (PBS; pH 8.0). Ultrasonic cell disruption was performed under these conditions: 300 W, working for 3 s, spacing 6 s, 90 cycles. After centrifugation at 12000 rpm for 30 min to remove precipitated protein and cell debris, the crude enzyme was loaded onto a Ni-NTA affinity column. After washing (washing buffer: 20 mmol L^−1^ Tris-Hcl, 500 mmol·L^−1^ Nacl, 40 mmol·L^−1^ imidazole, Ph 7.8) and eluting (elution buffer: 20 mmol·L^−1^ Tris-Hcl, 500 mmol·L^−1^ Nacl, 400 mmol·L^−1^ imidazole, pH 7.8), the purified proteins were eluted with elution buffer. And then the target enzyme from Ni-NTA purification was concentrated by ultrafiltration. Enzyme purity was analyzed by SDS-PAGE (10% separating and 5% stacking gels). Furthermore, the concentration of the purified enzymes was estimated using a modified Bradford protein assay kit (Sangon Biotech Co., Ltd. Shanghai, China).

### 4.5. Enzymatic Parameters of Wild-Type Enzyme and Mutant Enzymes

Specific activity: Added purified enzyme solution 20 μL (1 mg/mL) to substrate solution (0.2 mol/L HAc-NaAc buffer,40 μmol/L PLP, 50 mmol/L l-glutamate, pH 4.8), react at 48 °C for 10 min, and immediately put it into boiling water for 10 min to terminate the reaction. One unit of activity was equal to the amount of pure enzyme that produced 1 μmol GABA per min under the conditions described above. The specific activity is expressed as U/mg of protein. The GABA was derived with Dansyl chloride and then measured by HPLC. HPLC conditions were set as the references [43].

Kinetic Constants: Different concentrations (10–100 mmol/L) of substrate L-glutamate were prepared in 0.2 mol/L HAc-NaAc buffer (pH 4.8), and initial response rates were measured under different substrate concentrations.

Half-life of heat inactivation (t_1/2_): Wild-type enzyme or mutant enzyme was separately heat-treated for 0–140 min at 55 °C, and then the residual activity was measured. The activity of the corresponding enzyme without incubation at the same temperature was defined as 100%, and the relative activities of the enzymes were calculated. The thermal inactivation curve was fitted with Origin 8.0 software.

Semi-inactivation temperature (T_50_^15^): Wild-type enzyme or mutant enzyme were separately incubated in a 40–70 °C metal bath for 15 min, and then residual activity of the enzyme was determined. The activity of the enzyme at 40 °C after 15 min was taken as 100%, the relative activities data were fitted to the Boltzmann sigmoidal equation in Origin 8.0.

### 4.6. Molecular Dynamics Simulation of WT and its Mutants

In order to further explain the effect of the mutation site on the conformation of the protein, RMSD and RMSF in the molecular dynamics simulation trajectories of wild and mutant enzymes were calculated to compare the fluctuations of the potential energy for the whole system. In this study, the GAD crystal structure of *L. brevis CGMCC 1306* (PDB ID: 5GP4) was used as the template, and the three-dimensional structure visualization of the mutant enzyme was conducted by Pymol1.8. Molecular dynamics simulations of the WT and its mutant S325A were carried out in Gromacs2018.4 (Force field: AMBER99SB-ILDN). Each protein was soaked in a cubic box of simple point charge (TIP3P) water molecules. Sodium and chloride counterions were added into the system in order to preserve the electroneutrality and mimic the physiological environment. Each protein was subjected to a two-phase energy minimization. Then the MD simulation was performed for 10 ns at 300 K.

## Figures and Tables

**Figure 1 molecules-25-00690-f001:**
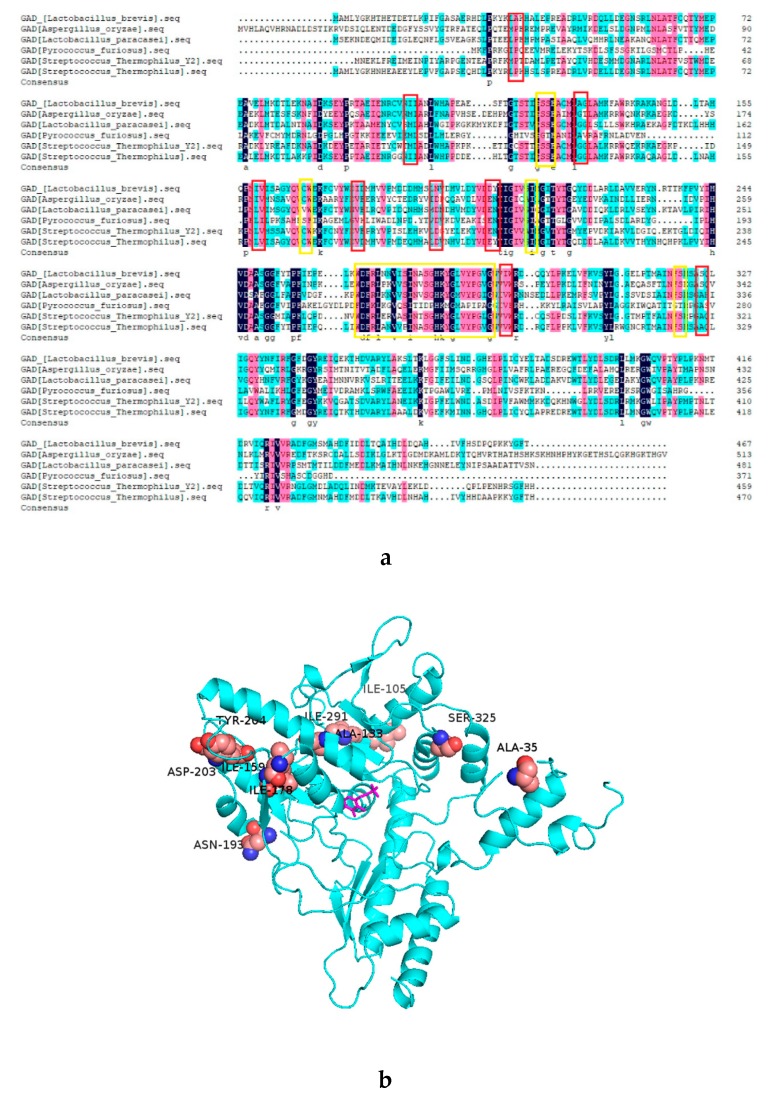
Putative important residues in glutamate decarboxylase identified by sequence analysis. (**a**) Sequence alignment of glutamate decarboxylase from L. brevis and other five glutamate decarboxylases (GADs). The amino acid sequences of GADs were aligned with the DNAMAN program using default parameters. The important residues of GAD are indicated with a yellow box. The consensus residues are indicated with a red box. (**b**) The consensus residues were visualized in the crystal structure of GAD (PDB ID:5GP4). Ten residues, Ala35, Ile105, Ala133, Ile159, Ile178, Asn193, Asp203, Tyr204, Arg225, and Ser325, are shown as spheres. The pyridoxal 5-phosphate (PLP) is shown as sticks.

**Figure 2 molecules-25-00690-f002:**
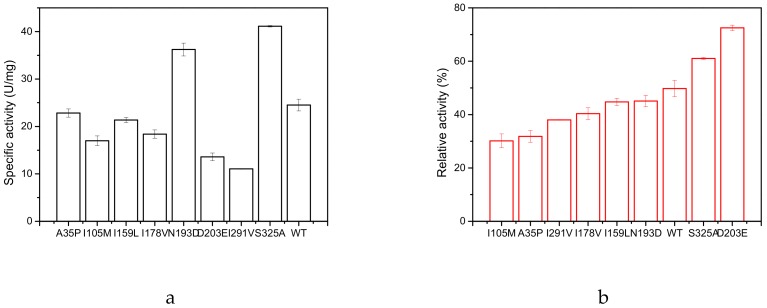
Specific activity and its inactivation profile of wild type (WT) and its mutants. (**a**) The specific activity of wild type and its mutants. (**b**) The residual activity of wild type and mutants after incubation at 60 °C for 20 min. The enzyme activity of each without heat treatment was assumed to be 100%.

**Figure 3 molecules-25-00690-f003:**
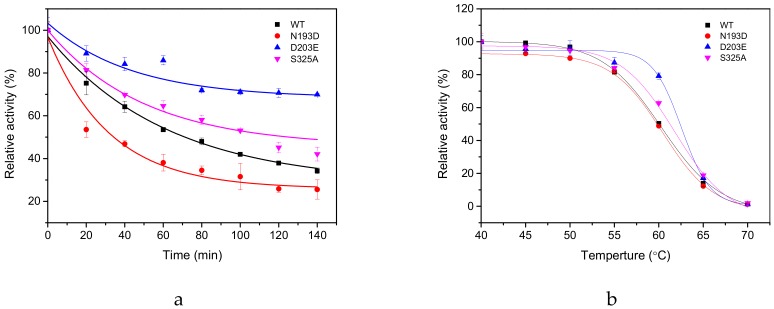
Stability analysis of WT and its mutants. (**a**) Thermal inactivation half-life (t_1/2_) of WT and its two mutants at 55 °C. (**b**) The thermal deactivation of wild-type GAD and the mutants at various temperatures for 15 min (T_50_
^15^). The enzyme activity of each without heat treatment was assumed to be 100%.

**Figure 4 molecules-25-00690-f004:**
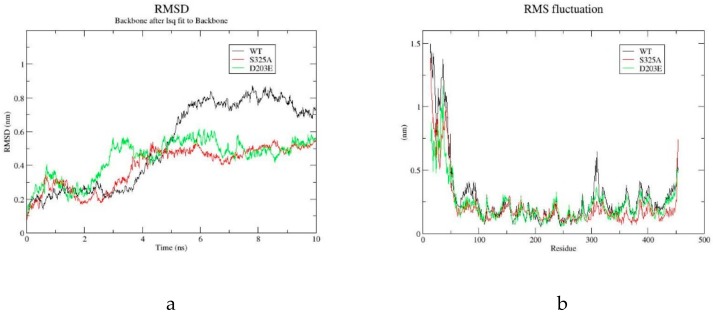
Molecular dynamics (MD) simulation analysis of WT and S325A for 10 ns at 300 K using Gromacs2018.4. (**a**) The root mean square deviation (RMSD) values of WT and its mutants D203E, S325A. (**b**) The root mean square fluctuation (RMSF) values of each amino acid during the simulation.

**Figure 5 molecules-25-00690-f005:**
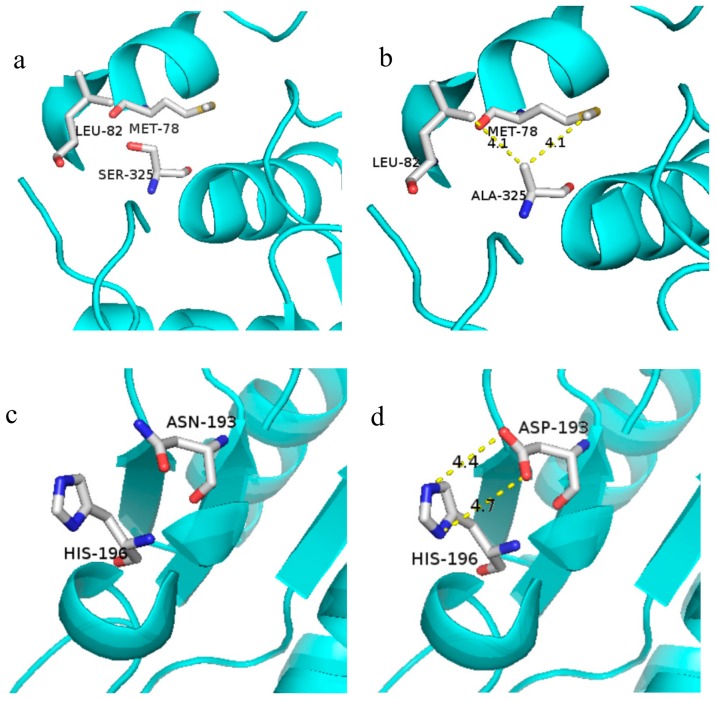
Molecular interaction simulation between the mutant enzyme (S325A, N193D) and WT. (**a**) The 3D structure diagram of Ser325. (**b**) In S325A, Ala325 was involved in hydrophobic interaction with the neighboring Met78 and Leu82. (**c**) The 3D structure diagram of Asn193. (**d**) In N193D, Asp193 was involved in an ionic bond with the neighboring His196.

**Table 1 molecules-25-00690-t001:** The ΔΔG _unfold_ (kcal·mol^−1^) of the 10 mutants.

Mutants	ΔΔG _unfold_T = 323 k	ΔΔG _unfold_T = 333 k	ΔΔG _unfold_T = 343 k	Mutants	ΔΔG _unfold_T = 323 k	ΔΔG _unfold_T = 333 k	ΔΔG _unfold_T = 343 k
D203E	−1.33	−1.34	−1.35	I178V	0.24	0.22	0.20
N193D	−0.86	−0.97	−1.13	I291V	0.84	0.81	0.79
S325A	−0.88	−0.88	−0.87	I159L	0.89	0.84	0.81
A35P	−0.76	−0.81	−0.82	A133G	1.59	1.61	1.63
I105M	−0.53	−0.53	−0.52	Y204N	2.17	2.22	2.19

**Table 2 molecules-25-00690-t002:** The specific activity and kinetic constants of wild type and its mutants.

Name	Specific Activity (U/mg)	*K_M_* (mM)	*kcat* (s-1)	*kcat/**K_M_* (s^−1^·mM^−1^)
N193D	36.28	34.35	137.56	4.00
D203E	13.64	26.60	40.21	1.51
S325A	41.12	42.39	176.19	4.16
WT	24.57	39.72	102.12	2.57

**Table 3 molecules-25-00690-t003:** Comparison of biochemical and kinetic properties of GADs from various sources.

Source	Optimum Temperature	Optimum pH	*K_M_* (mM)	Molecular Mass	Sequence Similarity
GAD(B1B389) [38] *Lactobacillus paracasei*	50 °C	5.0	5.0	57 kDa	48.2%
GAD(Q0GE18) [39] *Streptococcus salivarius ssp.* Thermophilus Y2	55 °C	4.0	2.3	52.6 kDa	46.5%
GAD(Q8U1P6) [40] *Pyrococcus furiosus*	75 °C	6.0	2.22	41 kDa	23.8%
GAD [41] *Aspergillus oryzae*	60 °C	5.5	13.0	48 KDa	43.0%
GAD [42] *Streptococcus thermophilus*	52 °C	4.2	5.0	53 KDa	72.1%

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
