# Peer review of "Parallel Strategy Increases the Thermostability and Activity of Glutamate Decarboxylase"

_molecules, 2020, doi:10.3390/molecules25030690_

Round 1
Reviewer 1 Report
The manuscript describes the production of an improved GAD variant for application in GABA biosynthesis. It is in general of scientific merit, however, there are some points that need to be altered.
These are my main remarks:
The introduction lacks references related to previous work on this particular GAD, as well as other homologous GAD enzymes, with an emphasis to their thermostability. Eg Page 1, lines 36-38: add a reference for the cloning and characterization of GAD.
Results, paragraph 2.1: the authors should indicate the aminoacids that are conserved and that are important for GAD catalytic activity, both in the text and in Fig.1A, including the appropriate references.
The red squares in Fig.1A are misplaced. The legend of Fig.1B should state which was the PDB used to make the figure.
According to the authors, “In homologous structures, the most frequently occurring amino acids at a particular position contribute to the stability more than other residues at the same position”. Based on this argument, it is not clear why the authors chose to mutate residues that were almost conserved in all thermostable homologs, such as I105M and A133G. Please explain.
Page 4, line 110: mutant N193D has a considerable increase in Kcat, so the overall improvement should not only be attributed to improved substrate binding efficiency.
In the “Discussion” section, and since the structure of GAD is known, the authors should support the analysis in page 6, lines 168-179, by appropriate figures showing the aminoacid interactions analysed in the text.
Materials and methods:
Paragraph 4.5, line 225: what is the substrate concentration in the assay?
Page 8, line 246: to my knowledge, PyMol is only used for visualizing protein structures and not for creating models.
Last, the language needs substantial improvement throughout the manuscript.
Minor corrections:
Page 1, line 32: explain “GABA”
Page 6, line 167: replace tryptophan by serine.
Page 7, line 200: add the PDB of the wild type GAD.
Reviewer 2 Report
The manuscript presented by Qing-Fei Zhang et al. describes the production by genetic manipulations and characterization of a thermostable mutant of glutamate decarboxylase from Lactobacillus brevis of high GABA-producing activity. The authors have used a particular approach of mutagenesis targeting conserved residues among homologous bacterial glutamate dehydrogenase and computer-generated Gibbs energy of protein folding for all the mutants generated. This combination of approaches is what the authors called a "parallel strategy", if this referee correctly understood. I have major concerns on this work that I therefore do not recommend for publication as such. A revised version could be proposed again for publication but only after major revisions have been applied to it in a revised version.
Major Concerns
It is impossible for the reader to understand the units of the different rates of reaction presented and discussed in this manuscript. In line 97, we read that the GAD mutant N193D shows a specific activity of 92.4 U per mg. In the M&M section, we read that unit U is "defined as the amount of GABA produced per milligram every minute". First, there is a total lack of indication on how the amounts (with the plural form here please) of GABA are measured with … moles? mmoles? nmoles ? µgrams? nanograms? The authors should introduce in which units they measure the amounts of GABA produced. Second, it is not indicated in this definition to what belong the milligrams stated in this definition … this referee assumes that these are mg of protein since the amount of GABA produced is explicitly defined in the text "per milligram" of something else then … The authors should precisely indicate milligram of protein. Und third, the unit U presented in this manuscript is self-explicit: an amount of GABA produced per mg of protein every minute, so to speak: per min. U is the unit of a specific activity. It is therefore false to write U/mg in the Results section as can be read in lines 97, 98. Moreover, the units of the specific activities in Figure 2 (left panel) are indicated to be %, i.e. a ratio. But the S325A bar of this histogram rises up to about 105, and in line 97, we read that the specific activity of the mutant S325A was indeed about 103 U/mg. SO there is an obvious error on the units on the legend of Figure 2a. Etc. Therefore, any independent reader, this referee included, cannot assess at all the weight of the discussion made by the authors from their results. The first person to present in a published paper the concept of directed in vitro evolution of an enzyme is NOT Frances Arnold but Willem P.C. Stemmer in 1994 (W.P. Stemmer. 1994. Rapid evolution of a protein in vitro by DNA shuffling. Nature, 370, 389-391). Willem Stemmer did not share the 2018 Nobel Prize in chemistry with Frances Arnold, George Smith, and Gregory Winter because he prematurely died of a cancer on April 2, 2013. The authors should change this sentence and reference accordingly. Adequate referencing is mandatory. Similarly, it would have been adequate to cite the first review paper published on glutamate decarboxylase by a prominent and pioneering scientist of this field: Sze P.Y. Glutamate decarboxylase. Adv Exp Med Biol 1979, 123, 59-78. Please add this reference to your list and cite it in the Introduction. Table 1 is clearly not comprehensive, references to a large number of older works on these topics are missing, but the text remains too elusive about this. For instance, well before 2018, a large number of researchers outside China have published data from sequence-based strategy to stabilize proteins. Could the authors explicitly state in their revised text that this table is NOT comprehensive and that others have made the same type of research? The authors should, for both design strategies, present at least the first paper published on this. Adequate referencing is mandatory. The sentence: "… the signature sequence of GAD was targeted (how? why?) by comparing (how?) consensus residues with these enzymes." as such is not understandable for the general reader. In particular, how do authors proceed here when comparing several consensus residues with whole enzymes? What does it mean? The authors should deepen their text in this part of the Results section to make it easily understandable for the reader. Shifting the KM from 34.35 up to 39.7 mM (lines 103-104) is a 15% increase, this is not experimentally significant. The authors thus cannot assert that the affinity of the mutant GAD was increased compared to the wild-type GAD. Moreover, the KM is NOT a binding affinity. This is a measure of the affinity (in a broad acceptance) in turn-over conditions. The binding affinity is measured by the experimental constant KD as being the ratio k-1 / k1, when the KM of a pure Michaelian enzyme is (k-1 + k2) / k1. The KM of a pure Michaelian enzyme presents a value close to the KD of the substrate ONLY WHEN k2 is largely inferior to k-1. The authors present no data showing that their enzyme is a pure Michaelian enzyme and no data showing that, in their enzyme, the microscopic rate of product formation is inferior to the rate of substrate dissociation. The authors thus cannot conclude what they conclude in their text. The thermal inactivation measured for N193D mutant and the wild-type GAD shows that at 60°C, the residual activity of these two enzymes after 20 min is about 45% for the mutant and about 50 % for the wild-type enzyme. In lines 122-123, we read that the thermal inactivations of the same two enzymes but now at 55 °C are characterized now by a t½ of about 33 min for N193D and of about 71 min for the wild-type on a 120-min incubation. There is a major discrepancy between these two sets of values, since in one case and at a higher temperature, N193D and WT GADs exhibit a highly similar thermostability whereas at a lower temperature (55°C) they mutant is much more rapidly inactivated than the WT enzyme. Could the authors comment on that apparent discrepancy? The mutant GADs D203E and S325A have a highly similar low RMSD, showing they are more rigid than the wild-type enzyme. Despite this, the mutants D203E and S325A have a completely different kinetic behavior toward glutamate, one (D203E) is poorly active when the second is more active than the wild-type enzyme. Can the authors comment on that apparent discrepancy? The authors state in line 167: "… tryptophan which cannot form hydrophobic interaction with its adjacent amino acid residues”. This is totally wrong, the amino acid tryptophan presents an indole side chain which forms ONLY hydrophobic interactions! Tryptophan is apolar and hydrophobic as phenylalanine is. The authors should change this whole paragraph accordingly.
Minor Concerns
Lines 39-40. There is a typo: "good active" should be "good activity". Line 43. There is a typo: "mimiced" should be "mimicked". Line 46. The assertion: "… has been understood better" requires reference(s). The authors should add them to a revised Introduction. Line 47. There is a double redundancy in this sentence, which reads: "… using a rational design strategy to design proteins rationally …" that might be changed in: "… using a rational design strategy to improve their thermostability…" Line 67. Please, the authors should state explicitly which algorithm they used to calculate the DDG of unfolding… and repeat this information, as it is in this manuscript, in the M&M section. Line 75. What is the exact meaning of a signature sequence? The use of the singular: "the signature sequence" implies that there is only one signature sequence in glutamate decarboxylases, is this correct? If not, change this sentence. Lines 82-83. The authors present 5 mutants of L. brevis glutamate decarboxylase as showing a DDG inferior to 0.5 and later in the same sentence, they state that they selected the first 8 mutants … Is it 5 or 8? Figure 1a. About the red box (not red "square") … they seem to not be properly positioned on the sequence alignment, a red box is even seen to highlight some void on the right of the figure at amino acid position 327. The authors are requested to correct that. In Figure 1b, the ligand should be colored differently that the protein backbone for sake of clarity. In Table 2, please indicate the units of the DDG values. Line 103. Km should be written KM at all occurrences in the text since the M refers to the name of real people, Mrs Menten or Mr. Michaelis. Please use a upper case M and not a lower case for KM. Line 122. The standard deviations on the t½s are missing. Line 129. RMSD and RMSF abbreviations are not defined. Line 130. The authors should state that the RMSD presented here is the global value of all carbon-alpha of the backbone of each wild-type and mutant GAD, if this referee is correct. Line 135. The authors link the smaller protein conformation shift they observed for D203E and S325A mutants to an improvement of the protein stability. To my opinion, there is no obvious link between the two. Could the authors comment on that and show references supporting their assertion? Line 153. In the sentence: "The first was ..." The first what? The closest word in the text that corresponds to this "first" is: "structure rigidity". Could the authors modify their sentences to the sake of clarity? Line 158. A linking word is missing. The part of the sentence: "Such results manifested combination…" should read: "Such results manifested that combination…" Lines 162-163. This sentence, as such, is not understandable. Its English is not correct. Lines 165-166. There is a part of the sentence that is missing between: "with distinct” und: "increases". As such, this sentence is not understandable. Its English is not correct. Lines 166-167. The "wild enzyme" should be written "wild-type enzyme". No enzyme turns wild in this world! The whole sentence is written in bad English and should be corrected, besides containing a false statement (See Major concern #8). Line 177. Typo error here: "a-helix" should be "alpha-helix". Alpha not a. Lines 180-182. The sentence is difficult to read and understand. Its English should be improved. Line 184. The word "mutational" here appears to not be located correctly. Discussion. The authors should discuss their own results at the light of what was done and published before by others such as: C. Jun et al. Thermostability of glutamate decarboxylase from E. coli … J Biotechnol 2014, 174, 22-28; L.-Q. Fan et al. Increasing thermal stability of glutamate decarboxylase… J Biotechnol 2018, 278, 1-9; E.S. Lee et al. Gene expression and characterization of thermostable glutamate decarboxylase … Biotechnol Bioprocess Eng 2013, 18, 375-381; N. Komatsuzaki et al. Characterization of glutamate decarboxylase from a high gamma-aminobutyric acid (GABA)-producer, Lactobacillus paracasei. 2008, 72, 278-285; Etc.). Table 4. Like elsewhere, KM and not Km. The column title "molecular weight" is not correct but should be molecular mass instead. The correct abbreviation for kilodalton is not kD but kDa. Lines 220-221, section Materials and Methods. The elution of the GAD from the Ni-NTA column is not described. Please precise te protocol used and the imidazole concentrations used for the different buffers, especially at which imidazole concentration the target enzyme is unloaded. Lines 224-225. Enzyme activity definition is not correctly stated here. The authors write: "the amount of GABA produced" but they do not state precisely the units of these amounts (mg ? grams? µg? mols? mmols? pmols?). Besides, "the amount" should be the plural: "the amounts". They state one step further: “the amount of GABA per milligram" without further indication. Is this milligram of what ? sugar ? protein? lipids?. And they finish this definition by: "every minute" which is not correct and should be "per min". Line 233. The sentence: "… was separately heat treatment for 120 min at 55°C and then calculated residual enzyme activity.” is bad English. A correct sentence could be: "… was separately heat-treated … and then the residual activity was measured." Line 234. "Each" ... each what? Line 237. The sentence: "… and then to determine the residual activity of the enzyme" is bad English. A correct sentence could be: "… and then residual activity of the enzyme was determined." In the legend of Figure 2, "wide type" is not correct English and should read: “wild-type”.Author Response
Please see the attachment.

Round 2
Reviewer 2 Report
In this revised manuscript, all my previous concerns (major and minor) have been properly addressed.